# Time series anomaly detection via Hypothesis testing for dynamical systems

## Abstract

Real world systems—such as robots, weather, energy systems and stock markets—are complicated and high-dimensional. Hence, without prior knowledge of the system dynamics, detecting or forecasting abnormal events from the sequential observations of the system is challenging. In this work, we address the problem caused by high-dimensionality via viewing time series anomaly detection as hypothesis testing on dynamical systems. This perspective can avoid the dimension of the problem from increasing linearly with time horizon, and naturally leads to a novel anomaly detection model, termed as DyAD (Dynamical system Anomaly Detection). Furthermore, as existing time-series anomaly detection algorithms are usually evaluated on relatively small datasets, we released a large-scale one on detecting battery failures in electric vehicles. We benchmarked several popular algorithms on both public datasets and our released new dataset. Our experiments demonstrated that our proposed model achieves state-of-the-art results.

## 1 Introduction

Hypothesis testing aims to decide whether the observed data supports or rejects a default belief known as the null hypothesis. Applications are abundant. In this work, we view anomaly detection as an application of hypothesis testing. This perspective is nothing profound—samples from the null hypothesis can be viewed as in-distribution, and rejection can be viewed as detecting anomalies. Despite being rather straightforward, this view was not carefully investigated in large-scale anomaly detection tasks, because most classical hypothesis testing methods suffer from the curse of dimensionality.

In this work, we address the problem incurred by high-dimensionality via focusing on *time series data collected from unknown dynamical systems*. We exploit the structure of dynamical systems and show that although the time series data can be high dimensional due to the long time horizon, the problem still remains tractable. More specifically, the concentration that leads to statistical confidence does not come from independent variables but from martingales. We turn the high dimensionality caused by the long time horizon into our favor. Furthermore, our analysis leads to a detection procedure in which the anomaly in systems (e.g., errors and attacks) can be isolated from the rarity of system input (e.g., control commands), and hence reduces misclassification rates.

By combining the above analysis with autoencoder-based probabilistic models, we develop a new model termed DyAD (DYnamical system Anomaly Detection). We show that the theory-motivated DyAD model can achieve state-of-the-art performances on public datasets including MSL (Mars Science Laboratory rover) (Hundman et al., 2018) and SMAP (Soil Moisture Active Passive satellite) (O'Neill et al., 2010). To further validate our finding, we then release a much larger (roughly 50 times in terms of data points) dataset to benchmark several popular baselines.

Our released dataset focuses on the battery safety problem in electric vehicles. In recent years, electric vehicle (EV) adoption rates increased exponentially due to their environmental friendliness, improved cruise range and reduced costs brought by onboard lithium batteries (Schmuch et al., 2018; Mauler et al., 2021). Yet, large-scale battery deployment can lead to unexpected fire incidents and product recalls (Deng et al., 2018). Hence, accurately evaluating the health status of EV batteries is crucial to the safety of drivers and passengers. To promote research in this field, we release a dataset collected from 301 electric vehicles recorded over 3 months to 3 years. Only battery-related data at charging stations was released for anonymity purposes. 50 of the 301 vehicles eventually suffered from battery

failure. Experiments on the EV battery dataset confirm that our proposed model achieves better performance for system anomaly detection.

In summary, our contributions are:

- We formulate hypothesis testing based on data observed from dynamical systems and derive generalized likelihood ratio test that exploits the Markovian structure of observations from dynamical systems.
- We show that the above formulation leads to a novel model, termed DyAD, for anomaly detection on dynamical systems.
- We release a large dataset collected from 301 electric vehicles, out of which 50 suffered from battery failure. In addition to benchmarking anomaly detection algorithms, the dataset may be of independent interest for machine learning tasks in nonlinear systems.

## 2    RELATED WORKS

### 2.1    ANOMALY DETECTION AND OUT-OF-DISTRIBUTION DETECTION

The difference between out-of-distribution detection (OOD) and anomaly detection (AD) is subtle. Up to the authors' knowledge, anomaly detection, as compared to OOD detection, refers to identifying samples that differ more drastically but rarely from the in-distribution samples. However, the mathematical formulations for the two problems are the same, and hence we use the two terms interchangeably within the context of this manuscript.

The core idea of AD is to develop a metric that differs drastically between normal and abnormal samples. Some previous works find that the model output probability for normal samples is higher (Hendrycks & Gimpel, 2016; Golan & El-Yaniv, 2018) in image tasks. Some previous works focus on detecting anomalies in the feature space by forcing/assuming the feature concentration of normal samples (Schölkopf et al., 1999; Lee et al., 2018). Some enhance the representation power of networks by introducing contrastive learning (Winkens et al., 2020; Tack et al., 2020) and data transformation (Golan & El-Yaniv, 2018). Ren et al. (2019) partition the input into semantic and background parts and define the log-likelihood difference between a normal model and a background model as the likelihood ratio to distinguish anomalies. More recently, Ristea et al. (2022) propose a self-supervised neural network composed of masked convolutional layers and channel attention modules for vision tasks, which predicts a masked region in the convolutional receptive field. Roth et al. (2022) utilize a memory bank learned from nominal samples and nearest neighborhood search to detect anomalies on industrial images.

### 2.2    TIME SERIES ANOMALY DETECTION

Since the battery system is a complex system that consists of multi-dimensional time series data, the most relevant deep learning research topic is multivariate time series anomaly detection. We will briefly introduce recent progress and the common datasets used in this area.

Several recent works focus on multivariate time series anomaly detection. Malhotra et al. (2016) propose to model reconstruction probabilities of the time series with an LSTM-based encoder-decoder network and use the reconstruction errors to detect anomalies. Hundman et al. (2018) leverage the prediction errors of the LSTM model to detect telemetry anomaly data. Su et al. (2019) propose OmniAnomaly to find the normal patterns through a stochastic recurrent neural network and use the reconstruction probabilities to determine anomalies. Zhao et al. (2020) capture multivariate correlations by considering each univariate series as an individual feature and including two graph attention layers to learn the dependencies of multivariate series in both temporal and feature dimensions. Deng & Hooi (2021) adopt graph neural networks to learn the inter-variable interactions.

There are several public time series datasets for anomaly detection. The SMAP (Soil Moisture Active Passive satellite) dataset is collected by a NASA's Earth Environment Satellite Observation Satellite (O'Neill et al., 2010). The MSL (Mars Science Laboratory rover) collects data sequences to determine if Mars was ever able to support microbial life (Hundman et al., 2018). The water

treatment physical test-bed datasets, SWaT (Secure Water Treatment) (Mathur & Tippenhauer, 2016), and WADI (Water Distribution) (Ahmed et al., 2017), are sensor data recording simulated attack scenarios of real-world water treatment plants. The TSA (Time Series Anomaly detection system) contains time series data collected from Flink (Zhao et al., 2020).

The anomaly labels in these time series datasets are marked when an anomaly event happens. In contrast, battery system failures can only be marked on the vehicle level rather than the event level. Therefore, though the mentioned multivariate time series anomaly detection algorithms achieve good performance on their respective datasets, the performance of these algorithms on our battery system failure detection dataset needs to be reassessed.

## 3  ANOMALY DETECTION FOR DYNAMICAL SYSTEMS

In this section, we start by formulating time series anomaly detection as hypothesis testing on dynamical systems. We then study likelihood-based tests in this context and exploit the Markovian structure of the time-series data collected from dynamical systems. By adapting classical analysis, we provide a guarantee for the false discovery rate of our model.

### 3.1  HYPOTHESIS TESTING AND ANOMALY DETECTION

Hypothesis testing on time series tries to decide whether a collection of random samples $\{x_1, x_2, ..., x_T\}$ provides enough evidence (in terms of statistical significance) to reject the null hypothesis that the samples come from a null distribution $p_0$ as opposed to a family of alternative distributions. In practice, the null distribution $p_0$ is often not known. To carry out standard hypothesis tests, $p_0$ needs to be estimated empirically from independent copies of random samples

$$\{x_1, x_2, ..., x_T\}_i, i = 1, 2, ..., n.$$

If the estimated $p_0$ is accurate enough, one can still apply standard hypothesis testing algorithms. To detect anomaly given a collection of time series $\{\mathbf{x}_i\}_i$, where $\mathbf{x}_i = \{x_1, x_2, ..., x_T\}_i$, the model can learn the normal distribution $p_0$. Then, given a new sample $\mathbf{x}$, the model predicts whether it is from the normal distribution (i.e., not rejects the null hypothesis) or from an abnormal distribution (i.e., rejects the null hypothesis).

The procedure is conceptually simple, yet the challenge lies in efficiently approximating the distribution $p_0$. In the following parts, we show how viewing $x_t$ as outputs from dynamical systems can relieve the curse of dimensionality.

### 3.2  HYPOTHESIS TESTING IN DYNAMICAL SYSTEMS

In this subsection, we formulate hypothesis testing in dynamical systems and highlight its difference against the classical setup. For simplicity, we consider a discrete dynamical system with a fixed time length in a Euclidean space. A more generalized and formal study is left as future directions. In particular, consider a random mapping

$$f : \mathcal{X} \times \theta \times \mathcal{U} \to \mathcal{X},$$

where $f$ describes the transition probability that maps an inner state $x_t \in \mathcal{X}$, a system input $u_t \in \mathcal{U}$, and a time-invariant system parameter $\theta \in \Theta$ to a **random** next state $x_{t+1} \in \mathcal{X}$. More formally, for $t = 1, 2, ..., T$,

$$x_{t+1} \sim f(x_t, \theta, u_t),$$

Furthermore, we consider the case when system inputs are sampled from a distribution that is independent from the system itself, $u_{1:T} \sim U$.

Our goal is to detect whether an observed sample comes from a normal system, where system parameters are sampled from the null hypothesis $H_0$, or from an abnormal system, where the parameters are sampled from the alternative hypothesis $H_1$:

$$H_0 : \theta \sim \Theta_0,$$
$$H_1 : \theta \sim \Theta_1, \text{for some } \Theta_1 \in \boldsymbol{\Theta}$$

The above formulation subsumes many real-world problems. For example, if we aim to detect abnormal electric vehicle batteries, $\theta$ can describe battery health, whereas the signals $x_t$ are recorded by the battery management system under the charging current $u_t$.

The benefit of viewing time series as observations from dynamical systems is that it turns the high dimensionality caused by the long time horizon into our favor. Intuitively, the problem dimension is determined by the dimension of the system dynamics $\theta$, whereas additional observations $x_t, u_t$ tell us more about the unobserved parameter $\theta$. We pursue this idea in the next subsection.

### 3.3 Likelihood tests for the dynamical systems

Following the notations above, the anomaly detection task can be stated via the optimization problem below:

$$\max_{c \in \mathcal{F}} \mathbb{E}_{u_{1:t} \sim U, \theta \sim \Theta_1}[\mathbb{I}\{c(x_{1:T}, u_{1:T}) = 1\}], \tag{1}$$

$$s.t \ \mathbb{E}_{u_{1:t} \sim U, \theta \sim \Theta_0}[\mathbb{I}\{c(x_{1:T}, u_{1:T}) = 1\}] \leq \alpha. \tag{2}$$

where, $\mathbb{I}$ denotes the indicator function, the hypothesis class $\mathcal{F}$ is a subset of prediction functions $\{c : \mathcal{Y}^{\otimes T} \times \mathcal{U}^{\otimes T} \to \{0, 1\}\}$, and the parameter $\alpha$ controls the false discover rate. Here, the objective aims to maximize the power of the test on the alternative hypothesis, i.e., the percentage of anomaly detected. For now, we assumed that the alternative hypothesis is simple and single-valued.

With the above goal in mind, we then have the proposition below.

**Proposition 1.** For any $\alpha$, the optimal solution to the optimization problem above (also known as the uniformly most powerful test) can be written as thresholding the conditional likelihood ratio below

$$c(x_{1:T}, u_{1:T}) = \mathbb{I}\left\{\prod_t \frac{p_1(x_t|u_{t-1}, x_{t-1})}{p_0(x_t|u_{t-1}, x_{t-1})} > c\right\}, \tag{3}$$

where $p_0$ denotes the likelihood under the null hypothesis, $p_1$ denotes the likelihood under the alternative hypothesis, and $c$ is chosen such that $E_{u_{1:t} \sim U, \theta \sim \Theta}[\mathbb{I}\{f(y_{1:T}|u_{1:T}) = 1\}] = \alpha$.

The proof is a direct application of the famous Neyman-Pearson theorem along with the fact that

$$\frac{p_1(x_{1:T}, u_{1:T})}{p_0(x_{1:T}, u_{1:T})} = \prod_t \frac{p_1(x_t|u_{t-1}, x_{t-1})}{p_0(x_t|u_{t-1}, x_{t-1})},$$

where we use the Bayes rule and the Markovian property of the dynamical system.

The interesting observations from the above theorem are twofold. First, the optimal detector is independent of the input distribution $u_{1:t} \sim U$, but only depends on the conditional distribution $p(x_{1:T}|u_{1:T})$. Second, we get a product form that resembles likelihood ratios for independent variables, which suggests that we may get stronger statistical significance from martingale-style concentration bounds.

In practice, very often the alternative hypothesis is composite instead of simple. In this setup, the uniformly most power test may not exist. Alternatively, we have the following guarantee on false discover rate:

**Proposition 2.** If for any $y, x, u$, there exists a $\theta$ such that $p_\theta(y|x, u) = 1$, then we have that

$$f(y_{1:T}, u_{1:T}) = \mathbb{I}\left\{\prod_t p_0(x_t|u_{t-1}, x_{t-1}) < c\right\},$$

is the generalized likelihood ratio test. Under the null hypothesis, the likelihood ratio within the indicator function converges in distribution to $\chi$-squared distribution with freedom $d$, where $d$ is the dimension of the parameter space.

The proof applies Wilk's theorem to the following equation:

$$\sup_{p_1} \frac{p_1(x_{1:T}, u_{1:T})}{p_0(x_{1:T}, u_{1:T})} = \prod_t \frac{1}{p_0(x_t|u_{t-1}, x_{t-1})}.$$

The above proposition suggests that, asymptotically, we could simply reject the hypothesis by thresholding the likelihood according to the $\chi^2$ test and control the false discovery rate. Since log-likelihood is monotonic in likelihood, it is equivalent to finding a classifier of the following form:

$$f(x_{1:T}, u_{1:T}) = \mathbb{I}\left\{ l(\theta) + \sum_{t \leq T} l(x_t | u_{t-1}, x_{t-1}, \theta) < c \right\}, \tag{4}$$

where we dropped the subscript and use $l(\theta, x, u) := \log p_0(\theta, x, u)$ to denote the log-likelihood under the distribution of the null hypothesis. The form in equation 4 motivates us to propose the anomaly detection algorithm in the next section.

## 4 DyAD: AUTO-ENCODER-BASED ANOMALY DETECTION MODEL

We have seen above that the key to anomaly detection via hypothesis testing is to learn the distribution of $\theta \sim \Theta_0$. We adopt the variational inference (as in variational autoencoder (Kingma & Welling, 2013), diffusion models (Ho et al., 2020), etc.) formulation for this task. In particular, we parameterize the family of likelihoods $\mathcal{L}$ via weights in neural networks. Then we want to identify the likelihood $l^* \in \mathcal{L}$ that minimizes the KL divergence between the empirical distribution $\hat{p}_0$ and the probability function $p_{l^*}$ induced by the learned likelihood $l^*$. For a more detailed discussion on the variational inference, we refer the readers to Section 2.2 in Kingma & Welling (2013). Hence, we get,

$$\min_{l \in \mathcal{L}} D_{KL}(\hat{p}_0, p_{l^*}) = \mathbb{E}_{\mathbf{x}_i, \mathbf{u}_i, \theta_i \sim \hat{p}_0}\left[ \log\left( \frac{p_0(\mathbf{x}_i, \mathbf{u}_i, \theta_i)}{p_{l^*}(\mathbf{x}_i, \mathbf{u}_i, \theta_i)} \right) \right],$$

where $\mathbf{u}_i, \mathbf{x}_i$ are shorthands for the $i_{th}$ sampled input and output sequences. We note that as the numerator is independent of $l$, we can equivalently solve

$$\max_{l \in \mathcal{L}} \mathbb{E}_{\mathbf{x}_i, \mathbf{u}_i, \theta_i \sim \mathcal{D}_{train}}\left[ l(\theta^i) + \sum_{t \leq T} l(x_t^i | u_{t-1}^i, x_{t-1}^i, \theta^i) \right],$$

where we used the fact that likelihood can be rewritten as products due to the Markovian structure.

However, the above problem cannot be solved, because in practice we can only observe the system inputs and outputs $\mathbf{u}_i, \mathbf{x}_i$, whereas the system parameter $\theta$ remains unknown. Hence, a natural fix is to infer $\theta$ from observed data.

$$\max_{(l_e, l_d) \in \mathcal{L}} \mathbb{E}_{\mathbf{x}_i, \mathbf{u}_i \sim \mathcal{D}_{train}} \mathbb{E}_{\theta_i' \sim l_e(\cdot | \mathbf{x}_i, \mathbf{u}_i)}\left[ \sum_{t \leq T} l_d(x_t^i | u_{t-1}^i, x_{t-1}^i, \theta_i') \right],$$

where we marginalize over the unobserved $\theta$ by learning a posterior distribution. In practice, we cannot optimize over the entire probability space. Hence we simplify the problem with the following approximations:

1. We assume the distribution of the system parameters $\theta$ can be reparameterized (e.g., through a neural network) as a multivariate Gaussian $\theta \sim \mathcal{N}(0, I)$.
2. We assume that $x_t$ is Gaussian conditioned on $u_{t-1}, x_{t-1}, \theta$, i.e.,
$$x_t \sim \mathcal{N}(\mu_d(u_{t-1}, x_{t-1}, \theta), \sigma_d^2 I),$$
   where $\mu_d$ is a function to be learned and $\sigma$ is a hyper-parameter.
3. We assume the posterior is also Gaussian, i.e.,
$$\theta \sim \mathcal{N}(\mu_e(u_{t-1}, y_{t-1}), \sigma_e^2(u_{t-1}, y_{t-1})),$$
   where $\mu_e$ and $\sigma$ are functions to be learned.

Putting everything together and use the ELBO (evidence lower bound) trick, we get something that is similar to the LSTM-autoencoder model, but with one key difference, that the system input is sent into the decoder. The optimization problem now becomes:

$$\min_{\mu_d, \mu_e, \sigma_e} \mathbb{E}_{\mathbf{x}_i, \mathbf{u}_i \sim \mathcal{D}_{train}, \theta \sim \mathcal{N}(\mu_e, \sigma_e^2 I)}\left[ D_{KL}(\mathcal{N}(\mu_e, \sigma_e^2), \mathcal{N}(0, I)) + \sum_{t \leq T} (y_t^i - \mu_d)^2 / \sigma_d^2 \right]$$

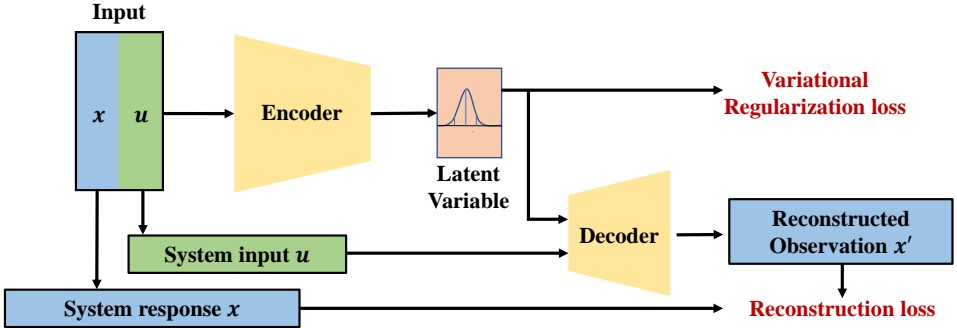

Figure 1: The network architecture of DyAD. The input data are split into system input $u$ and system response $x$. The model parameters are updated by minimizing an autoencoder style loss in Eqn. 5.

where $D_{KL}$ denotes the KL divergence, we take the negative sign and use that log-likelihood of Gaussian is quadratic. The notations $\mu_d$, $\mu_e$ are parameterized by recurrent neural networks and are shorthands for

$$\mu_e = \mu_e(\mathbf{x}_i, \mathbf{u}_i), \sigma_e^2 = \sigma_e^2(\mathbf{x}_i, \mathbf{u}_i),$$
$$\mu_d = \mu_d(u_{t-1}^i, x_{t-1}^i, \theta).$$

We further simplify the KL divergence using properties of Gaussian distributions and get

$$\min_{\mu_d, \mu_e, \sigma_e} \mathbb{E}_{\mathbf{x}_i, \mathbf{u}_i \sim \mathcal{D}_{train}, \theta \sim \mathcal{N}(\mu_e, \sigma_e^2 I)} \left[ \underbrace{\|\mu_e\|^2 + tr(\sigma_e^2) - \log(|\sigma_e^2|)}_{\text{variational reg. loss}} + \underbrace{\sum_{t \leq T}(y_t^i - \mu_d)^2/\sigma_d^2}_{\text{recon. loss}} \right] \quad (5)$$

If we view $\mu_e, \sigma_e$ as output of the encoder network and $\mu_d$ as output of the encoder network, then we can adopt the variational autoencoder training procedure with the modification specified in the Figure 1. The key difference is that, instead of requiring the network to learn to retrieve all data dimensions, our model aims to learn the internal states $\theta$ of the system. Therefore, the decoder in DyAD, which now serves as a dynamical system, is responsible for simulating the system, and retrieving the response from the latent representation and the internal states. We note that some recent works (e.g., Girin et al. (2020); Mehta et al. (2021)) also applied autoencoders to learning dynamical systems, yet our result provides the first derivation and application via anomaly detection.

We briefly comment on why our proposed algorithm could outperform existing methods before moving on to testing the empirical performance of DyAD. Existing deep-learning based methods mostly build on the intuition that a model could learn generalizable normal patterns in the time series. Our analysis formalizes this intuition with Proposition 1 and 2, from which we can make the following improvements. First, by proposing a dynamical system probabilistic model, we isolate rare input signals (e.g., battery charging protocol) from abnormal systems (e.g., battery status). Second, by focusing on hypothesis testing in the system parameter space $\theta$, the dimension of the distribution is significantly reduced. These two changes allow our model to utilize data more efficiently.

## 5 BATTERY SYSTEM ANOMALY DETECTION

Before we present the anomaly detection results, we briefly describe our released EV dataset. As one of the core components of EVs, the battery system not only needs to support long cruising range, but also guarantee the safety of drivers and passengers. Effective and efficient detection of battery system failure reduces the product recall rate and improves the user experience. Due to its wide range of applications and great significance, we release a large-scale battery dataset, which contains over 88M battery charging time steps collected from 301 vehicles (see detailed stats in Table 1). The multi-dimensional time series features include current, voltage, temperature and SOC (state of charge) information. An example is shown in Figure 2(b) and 2(c). More examples

|                        | MSL      | SMAP     | SWaT    | EV (ours)    |
| ---------------------- | -------- | -------- | ------- | ------------ |
| Number of dimensions   | 55       | 25       | 50      | 8            |
| Number of time frames  | 132, 046 | 562, 800 | 92, 255 | 88, 135, 040 |
| Anomaly ratio          | 10.27%   | 13.13%   | 5.95%   | N/A          |

Table 1: The statistics of our EV Dataset and public datasets.

can be found in Appendix B. Our dataset and code are available at `https://1drv.ms/u/s!AnE8BfHe3IOlg13v2ltV0eP1-AgP?e=n4oBM1`. The usage of our data is under CC BY-NC-SA license.

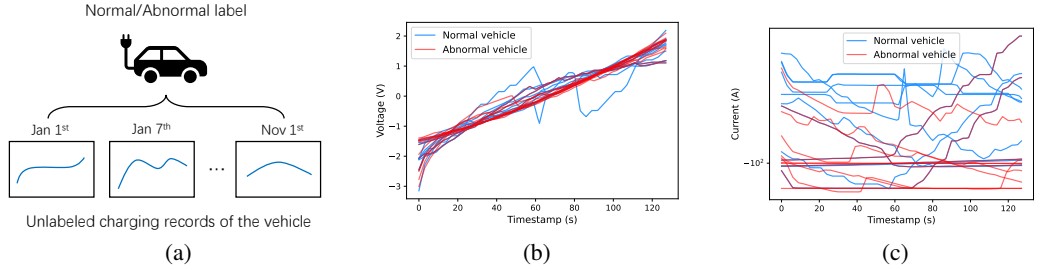

Figure 2: An illustration of our EV dataset. (a) The data structure of our EV dataset. Only vehicle-level anomaly labels are available. Charging snippets are collected within months and years. (b)-(c) The voltage and current of charging snippets collected from a normal and an abnormal vehicle.

We highlight that unlike previous datasets where anomalies are marked when an unexpected event happens in time series, the anomaly labels in the battery dataset have a natural hierarchical structure as shown in Figure 2(a). Specifically, abnormalities in the battery system are chronic (such as aging), so the abnormality labels are at the vehicle level (one per vehicle) rather than the event level (one per timestamp), which is a more challenging time series anomaly detection task. To address the problem, we propose the following procedure to summarize piece-wise prediction into vehicle-level prediction.

## 5.1 DETECT VEHICLE LEVEL ANOMALIES

To handle the hierarchical dataset structure and sparsity of anomaly labels, we develop a robust scoring function to map the snippet-level scores obtained from DyAD to vehicle-level predictions. The scoring function is shown in Alg. 1. In particular, we predict the abnormal degree of a charging snippet by thresholding the reconstruction error at value $\tau$ and then predict whether a vehicle is abnormal by averaging the top $h$ percentile errors. Both $\tau$ and $h$ are fine-tuned on the validation dataset. Notice that the straightforward idea is to average all charging snippets belonging to a vehicle, but we will show that our robust scoring procedure achieves better performance in experiments. We apply both the robust scoring function and the averaging scoring function to all deep learning baselines.

---

**Algorithm 1** Pseudo code of the robust scoring function

---

 **Hyperparameters:** percentile $h$ and threshold $\tau$.
 **Input:** Charging snippet scores $\vec{r} = \{(r_i)\}_i$ from a vehicle.
 **Output:** Vehicle-level prediction.
 1: Sort $\vec{r}$ by $r_i$ from large to small.
 2: Take the mean of the largest $h\%$ as the vehicle's score.
 3: Predict a vehicle as abnormal if the average of the largest $h\%$ is greater than $\tau$.

---

## 5.2 EXPERIMENT RESULTS

The interpolated averaged ROC curves and the mean and variance of AUROC values are shown in Figure 3 and Table 2, respectively. Our proposed algorithm DyAD achieves the best performance

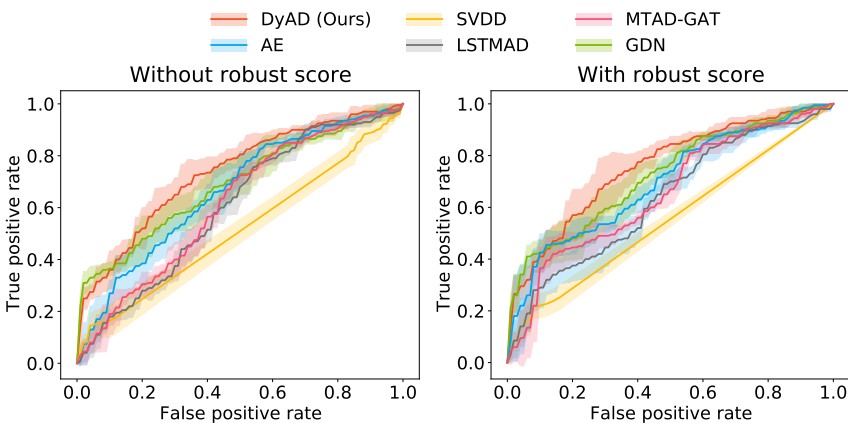

Figure 3: Interpolated averaged ROC curves of several algorithms on our EV battery dataset. Shaded area represents the five-fold variance.

on our large-scale dataset against well-established autoencoder-based and graph-based algorithms. Meanwhile, this is the first time to deploy deep learning algorithms to detect electric vehicle battery system failure in such a large dataset. Furthermore, DyAD benefits from the proposed robust scoring function with a 2.0% improvement. It suggests that sometimes averaging on all snippets belongs to a vehicle is not the best choice for such a hierarchical dataset, which is one of the difference between our dataset and traditional time series anomaly detection datasets.

Table 2: Mean and standard variance of *test* AUROC (%) values on all vehicles from three makes. Among all the considered algorithms, DyAD achieves the best detection results by a $3.1\% \sim 20.0\%$ AUROC boost and a relatively small variance compared to the second best algorithm on both averaging (**Left**) and robust scoring (**Right**) criteria. Further, our robust scoring function consistently improves baseline algorithms. Bold denotes the best results.

| Algorithm | AUROC (%) by Averaging Score | AUROC (%) by Robust Score |
|---|---|---|
| AE | $66.1 \pm 3.1$ | $69.4 \pm 2.6$ |
| Deep SVDD | $51.3 \pm 4.2$ | $55.2 \pm 2.5$ |
| LSTMAD | $60.0 \pm 1.5$ | $63.1 \pm 1.7$ |
| MTAD-GAT | $61.3 \pm 0.9$ | $65.2 \pm 2.3$ |
| GDN | $68.9 \pm 3.1$ | $72.1 \pm 3.2$ |
| **DyAD** (Ours) | $\mathbf{73.2 \pm 2.5}$ | $\mathbf{75.2 \pm 2.7}$ |

Table 3: Mean and standard variance of *test* AUROC (%) values on each make. Bold denotes the best results.

| Algorithm | Make 1 | Make 2 |
|---|---|---|
| AE | $65.3 \pm 7.0$ | $59.4 \pm 8.4$ |
| Deep SVDD | $51.6 \pm 5.7$ | $58.5 \pm 13.7$ |
| LSTMAD | $56.0 \pm 7.3$ | $58.6 \pm 9.5$ |
| MTAD-GAT | $57.1 \pm 9.9$ | $50.9 \pm 12.5$ |
| GDN | $64.8 \pm 11.5$ | $65.3 \pm 4.3$ |
| **DyAD+Robust Score** (Ours) | $\mathbf{78.0 \pm 5.2}$ | $\mathbf{84.4 \pm 3.8}$ |

We further examine the difficulty of detecting anomalies in each make by *training models on data from the same make only*. Make 3 is omitted since it has only four anomaly vehicles thus can not do the five-fold cross-validation. The AUROC values of different algorithms are shown in Table 3. It seems that learning one make of vehicles individually would lead to a larger variance of the results. One possible reason is due to the smaller amount of data. When the variance of the results becomes significantly larger, the mean value of the results becomes less reliable in comparing different algorithms affected by outlier values. Therefore, we recommend using a larger number of vehicles to reflect the performance of the algorithms in the population. In all five algorithms, the

variance of DyAD, although also increased, is still at a lower level than the other four. Meanwhile, the change of the other four algorithms is within variance. Based on the results in Table 3, detecting anomalous vehicles from Make 2 appears to be simpler than that from Make 1.

By comparing Table 3 against Table 2, we notice that some algorithms(e.g., GDN) benefit from training on all vehicles altogether, whereas some (e.g., DyAD) benefit from learning different models for each Make. Hence, there is still room for researchers to improve the detection of electric vehicle battery system failure. Limited to time and computing resources, we only implement these algorithms. We also welcome researchers to further develop algorithms on the dataset to improve EV safety.

## 6    TIME SERIES ANOMALY DETECTION IN OTHER REAL-WORLD TASKS

To supplement our results, we implement DyAD on two spacecraft system time-series anomaly detection datasets, MSL (Mars Science Laboratory rover) and SMAP (Soil Moisture Active Passive satellite), and a water treatment test-bed system dataset, SWaT (Secure Water Treatment). The two spacecraft datasets are released by NASA (O'Neill et al., 2010; Hundman et al., 2018), recording the spacecraft's telemetry channel sequences and command sequences encoded as 0 or 1. The water treatment dataset integrates digital and physical elements to control and monitor system behaviors. They are widely used as benchmarks for anomaly detection (Hundman et al., 2018; Zhao et al., 2020; Chen et al., 2022; Deng & Hooi, 2021). For these three datasets, the anomaly events are defined on time frames by another log. And the commonly used metrics are F1 score, Precision, and Recall.

We implement the same baseline algorithms with necessary modification for dataset dimension. The detection results are shown in Table 4. For the spacecraft system, we treat the command dimensions as system input and the telemetry data as system response. More training details can be found in Appendix C. As shown in the table, DyAD achieves the best F1 scores consistently. Specifically, we improve the SOTA algorithm by 3.9% and 3.0% on the MSL and SMAP datasets respectively. For the water treatment system, the sensor dimensions are treated as system input and the actuator dimensions as system response. More details can be found in Appendix C. We can see from the table that DyAD outperforms other baseline algorithms by 0.4%.

Table 4: F1, Precision and Recall on the two spacecraft datasets. Bold denotes the best results.

| Method | MSL | | | SMAP | | | SWaT | | |
|---|---|---|---|---|---|---|---|---|---|
| | F1 | Precison | Recall | F1 | Precision | Recall | F1 | Precision | Recall |
| AE | 0.5774 | 0.5507 | 0.6070 | 0.6215 | 0.8606 | 0.4864 | 0.7961 | 0.9452 | 0.6876 |
| Deep SVDD | 0.6804 | 0.8270 | 0.5779 | 0.2965 | 0.1752 | 0.9630 | 0.7870 | 0.8633 | 0.7231 |
| LSTMAD | 0.4502 | 0.3831 | 0.5458 | 0.6944 | 0.8914 | 0.5687 | 0.8007 | 0.9570 | 0.6883 |
| MTAD-GAT | 0.9084 | 0.8754 | 0.9440 | 0.9013 | 0.8906 | 0.9123 | 0.8359 | 0.9271 | 0.7612 |
| GDN | 0.7660 | 0.7598 | 0.7723 | 0.7069 | 0.7556 | 0.6641 | 0.8082 | 0.9935 | 0.6812 |
| **DyAD** (Ours) | **0.9438** | 0.9215 | 0.9673 | **0.9313** | 0.9501 | 0.9132 | **0.8399** | 0.9824 | 0.7335 |

## 7    CONCLUSIONS AND DISCUSSIONS

In this work, we have seen that properly formulating the probabilistic model underlying the data—in our case, viewing time series as outputs from dynamical systems— can greatly improve sample efficiency and lead to better model design. Our analysis naturally motivates a variant of the autoencoder model that is tailored for encoding dynamical systems. The performance of our proposed model is validated on both our released large-scale EV dataset and on existing public datasets.

There is much to be done. It is known that models based on likelihood alone do not achieve the best performances, as observed in Nalisnick et al. (2018). The idea is that although likelihood provides a guarantee of the false discovery rate, it does not necessarily give the best detection power, which would depend on the distribution of the alternative hypothesis. For this reason, multiple improvements (Ren et al., 2019; Xiao et al., 2020; Choi et al., 2018) have been made to further boost detection performances by improving the scores generated by likelihoods. As these investigations are orthogonal to our focus within this work, it remains to see whether similar ideas can be applied to improve our models.

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

## A    RELATED WORK ON BATTERY SAFETY

To the best of our knowledge, there is no well-established deep learning study of battery system failure with large-scale datasets. On one hand, deep learning technology has not been widely used in detecting battery failure, and many studies still use statistical methods (Xue et al., 2021) or canonical machine learning (Zheng et al., 2020). On the other hand, the absence of large public datasets hinders the progress of deep learning techniques in this area.

Existing researches mainly study lithium battery safety through battery physical/chemical structure and charging/discharging data. Physic-based approaches usually aim to improve battery safety by changing/adding the structure/component inside a battery. Wu et al. (2014) add a bifunctional separator inside the battery to achieve early detection of lithium dendrites. Yang et al. (2021) design a thermally modulated LFP (lithium iron phosphate) battery to improve cruise range. Advanced data-based methods analyze battery faults with deep learning models. Hong et al. (2019) adopt an LSTM network to predict multi-forward-step voltage value and judges the battery safety with a threshold voltage value. Li et al. (2020) propose to achieve higher battery fault diagnosis reliability and accuracy by combining the LSTM model and the equivalent circuit model. Yang et al. (2020) employ multi-layer neural network to estimate the current of the short-circuited cell and then predict maximum temperature increase with a 3D electro-thermal coupling model.

Meanwhile, although deep learning is a data-hungery technique, existing related studies do not incorporate enough vehicles in their datasets. We notice that Li et al. (2020) collects data from 9 vehicles and divides them into three categories, Hong et al. (2019) record time series information of one electric taxi, Zheng et al. (2020) use four cells with inconsistent capacities, and Yang et al. (2020) use eight cells. However, they cannot be used as a benchmark for measuring battery system anomalies because we believe that a conclusive result requires validation on a much larger dataset beyond several vehicles and their inconsistent data form is another problem.

## B    DATASET EXAMPLES

Some charging snippets from the battery dataset are depicted in Figure 4. The feature dimensions include voltage, current, max/min single cell voltage and max/min cell temperature. From the figure, we can see that the time series are mixed, and there is no simple rule to distinguish the charging snippets of abnormal vehicles from normal charging snippets. This is also one of the main reasons why anomaly labels are on the vehicle level rather than the snippet level.

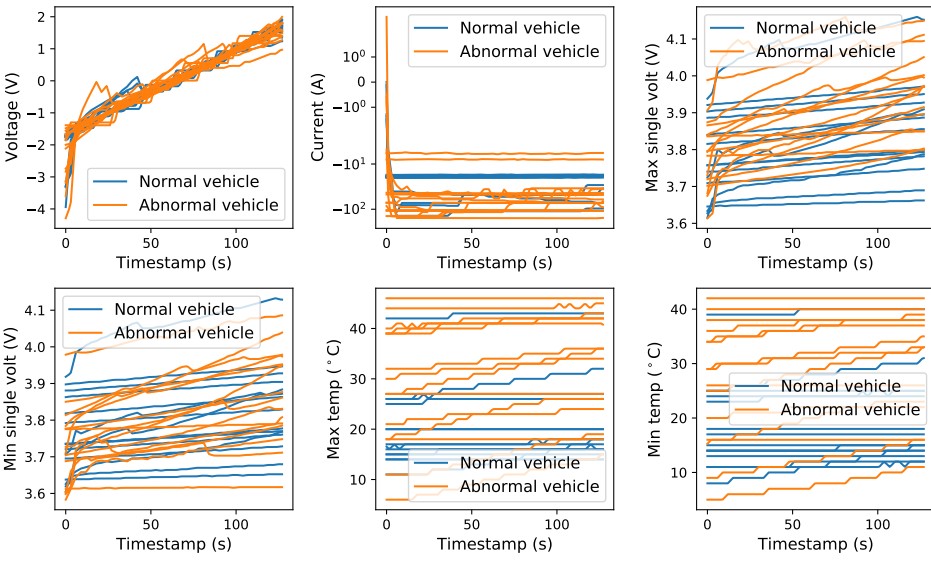

Figure 4: Charging snippet examples from snippets of normal and abnormal vehicles.

Here we also give a conceptual figure to help to understand what an anomaly in battery system would be like. In Figure 5, there are three charging snippets collected from a vehicle, where the first two have a normal max temperature value and the third shows a sharp increase in temperature. Notice that the different snippets are not contiguous, and there are discharging phase between snippets. Our dataset does not contain the third type of snippets, as our goal is to detect battery system failures early enough to prevent potential hazard in advance.

This example also explains that the labels in the dataset can only be labeled at the vehicle level, but not at the charging snippet level, since even a battery system that is about to fail may exhibit a normal charging pattern early on.

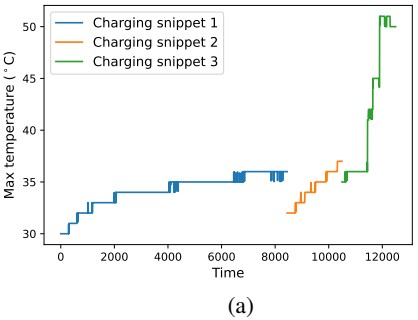
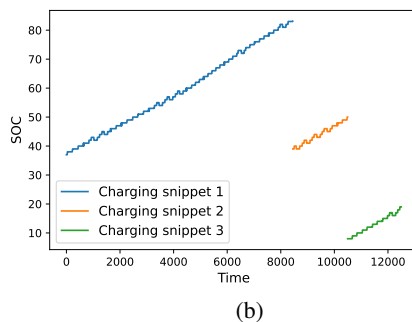

(a)                              (b)

Figure 5: An example of the anomaly in the EV dataset. (a) The max temperature of a vehicle with three charging snippets. (b) The SOC of the battery system with three charging snippets. The third snippet shows a runaway charging temperature, which means that the battery system is already broken.

## C    ALGORITHMS IMPLEMENTATION DETAILS

The ultra implementation details can be found in our released code. We partially use the official and public code resources from PyOD[1], LSTMAD[2], GDN[3] and MTAD-GAT[4]. All the experiments are run on a machine with four 2080 Ti GPUs.

### C.1    AUTOENCODER

The network adopts an encoder-decoder structure with batch normalization layers, drop out layers and sigmoid activation functions. It is built with several fully connected layers. The latent dimensions are [64, 32, 32, 64] in the encoder and decoder. We train the network 20 epochs with a batch size of 128 for the battery dataset, and 5 epoch for the two spacecraft datasets. We use the Adam optimizer with a learning rate of 0.001.

### C.2    DEEP SVDD

The feature extraction network of the deep SVDD is an autoencoder with hidden dimensions [64, 32, 32, 64], which is similar to the autoencoder above. The SVDD loss is computed on the middle 32-dimensional latent feature. We also adopt the reconstruction loss to help the network learn better data representation. We train 10 epochs with a batch size of 64 using the Adam optimizer with a learning rate of 0.001 for both the battery dataset and the spacecraft datasets.

---

[1]https://github.com/yzhao062/pyod
[2]https://github.com/PyLink88/Recurrent-Autoencoder
[3]https://github.com/d-ailin/GDN
[4]https://github.com/ML4ITS/mtad-gat-pytorch

### C.3    LSTMAD

The LSTMAD uses the LSTM layers and LSTM cells to encode and decode the input data, respectively. The latent feature dimension is 32. We use an Adam optimizer with a learning rate of 0.001 and train 20 epochs with a batch size of 128 on both the battery dataset and the spacecraft datasets. For the two spacecraft datasets, the window length is set to 128. The reconstruction loss function is the mean absolute error.

### C.4    MTAD-GAT

For the spacecraft dataset, the detection results are directly from the official paper Zhao et al. (2020). For the battery dataset, we set the window length to 100 and train the graph network 30 epochs with a batch size of 256. Only the feature dimension number is modified. The other parameters are set to the default value.

### C.5    GDN

We adapt the graph layers to fit our input data. Stacked fully-connected layers are attached at the end of the graph layer. The latent dimension is 128. The window length of the time series data is set to 32 for the battery dataset and 128 for the spacecraft datasets. We use an Adam optimizer with a learning rate of 0.001 to train 20 epochs with a batch size of 128.

### C.6    DYAD

For the battery dataset, we use GRU as the recurrent unit and train the network 3 epochs with a cosine annealing Adam optimizer. The hidden size and latent size are set to 64 and 32, respectively. The spacecraft dataset is pre-processed following Zhao et al. (2020). Different snippets are concatenated together. To deal with this special data structure, we use LSTM as the recurrent cell and add a convolutional layer to the decoder. A delayed signal of the encoder is passed to the decoder to compensate for sparse command channels. We train the MSL dataset for 10 epochs and the SMAP dataset for 5 epochs. Both with a cosine annealing Adam optimizer.

The SWaT dataset provides an official document interpreting each dimension of the data, including the type (sensor or actuator) and a brief description for each dimension. We treat the sensor dimensions as system input and the actuator dimension as system response. We directly apply DyAD on SWaT with a window size of 128.

We notice that another water treatment dataset, WADI (Ahmed et al., 2017), is also used in recent works (Deng & Hooi, 2021). However, as shown on the official dataset website, the current version of WADI has 127 dimensions, while the reported number of dimensions is 112 in recent research works (Deng & Hooi, 2021). Because of the version mismatch, we do not consider using WADI in our experiments.

