# OpenReview forum: "Time Series Anomaly Detection via Hypothesis Testing for Dynamical Systems"
_ICLR.cc/2023/Conference — Submitted to ICLR 2023_

### Official Review · Reviewer_4Vb4 · 2022-10-24

**Confidence:** 2
**Correctness:** 3
**Technical Novelty And Significance:** 2
**Empirical Novelty And Significance:** 3
**Recommendation:** 5

**Clarity, Quality, Novelty And Reproducibility:**

The assumptions and overall steps included in the constructoin of DyAD appear to be correct. As highlighted earlier, I think the paper’s overall writing could be improved in order for certain concepts to be more easily understood, especially the links between hypothesis testing and the overall DyAD architecture. Given how central the Battery Safety dataset is to the paper, I would also advise the authors to consider including more samples or illustrations that convey why the model architecture advised for in this work is especially suitable.

**Strength And Weaknesses:**

- I appreciate that the authors have introduced a new complex dataset for anomaly detection, whose properties may overlap with time series encountered in other real-world domains. I believe that moving beyond the standard datasets used for evaluation is a particularly effective means of identifying new research problems and directions.
- However, the paper treads a fine line between tackling an application-specific problem and proposing a more general solution to anomaly detection in multivariate time series datasets, which can be confusing at times. I believe the Battery System problem itself could be better explained early on in the paper via more illustrative examples of what a fault looks like. One other thing I was unsure of while reading the paper is whether the problem set-up of detecting anomalous vehicles (here described by several collected time series), would be better posed as a classification problem overall rather than time series anomaly detection. Given that the detection appears to be carried out offline, it is unclear whether there may be other more obvious means of detecting issues or degradations in this case.
- I found that there was a bit of a disconnect between sections 3 and 4, where it wasn’t always easy to understand how certain aspects introduced in Section 3 are then manifested in the architecture of DyAD.
- The use of auto-encoders for time series anomaly detection is fairly common, and I would have liked the novel aspects of this formulation to be more clearly expressed. The takeaways on why DyAD is more appealing than competing techniques was not clear to me by the end of reading the paper beyond the performance improvements obtained in the experiments.


**Summary Of The Paper:**

In this work, the authors propose a novel anomaly detection scheme based on hypothesis testing, that is directed towards multivariate problems where higher dimensionality can prove to be challenging for statistical testing. The authors leverage an autoencoder architecture for modelling the underlying dynamical system, and come up with a robust approach for determining whether a time series is anomalous or not. The contributions of the paper are framed around a new dataset on the battery power of electric vehicles that was curated for this paper (and made publicly available), but the authors also feature additional experiments on more widely-used datasets such as SMAP. Notable performance improvements are observed across all settings.

**Summary Of The Review:**

This paper frames its contributions around a novel dataset that poses different challenges to other anomaly detection datasets that are usually considered in the literature. The authors also develop a new model (DyAD) and robust scoring procedure that outperforms several other competing techniques. Nevertheless, I believe that some of the concepts put forward in the paper are not developed clearly enough, and the novelty of the modelling contributions were particularly unclear while reading the paper. Overall, I would encourage the authors to revise how the paper is currently structured in order to reduce confusion in regards to these aspects, and highlight the contributions more clearly.

---

> ### Author Response · Authors · 2022-11-14
> **Response to Reviewer 4Vb4**
>
> Thank you for your time reviewing our paper and for your valuable comments! We have edited the manuscript as suggested and believe that the comments improved our presentation. We hope to highlight the following points,
> 1. We added a few plots to illustrate the structure and content of the EV data. These plots were in the appendix before due to limited space.
> 2. We added explanations as suggested at the beginning and the end of section 4. We explain how the analyses in section 3 could lead to the architecture in section 4, as well as on why our model design can improve over vanilla VAE. In short, our model detects the abnormality in the input-to-output mappings, rather than the time series themselves. This can avoid obfuscation caused by rare inputs and can also reduce the dimension problem.
>
>
> Our study was motivated by the failure of existing methods on the EV dataset, yet our proposed analysis and framework can be applied to standard time series anomaly detection. We demonstrated this on two spacecraft datasets and added an additional water-system dataset. We believe that our theory framework (which we failed to find in other deep anomaly-detection work) and the new large-scale real-world dataset could be of the community’s benefit, and inspire new algorithm design.
>
> We are happy to answer any further questions.
>
> More details are as follows.
>
> **Q**: An example of what a fault looks like.
> **A**: We thank the reviewer for this suggestion. Now we add plots in the main text and appendix B, showing how the anomaly faults look like and the hierarchical dataset structure (anomalies can only be labeled on the vehicle level rather than the snippet level). We comment that EV abnormal detection focus on predicting problems **weeks ahead** of the battery incidents to allow time for inspection. Therefore, data records of the incidents were removed, and humans cannot easily judge the health of batteries by simply looking at the charging patterns.
>
>
> **Q**: The paper treads a fine line between tackling an application-specific problem and proposing a more general solution?
> **A**: The EV battery system anomaly detection is more challenging since the anomaly state is not directly observable during daily usage and the anomaly labels are very sparse (see the conceptual example in Appendix B for better understanding). We start by tackling this particular challenging problem, but we find existing methods fail to give reasonable predictions. To solve the problem, we exploit the time series structure to propose a general solution that not only solves the EV battery system anomalies but also improves performance on existing datasets.
>
>
> **Q**: The problem would be better posed as a classification problem rather than time series anomaly detection?
> **A**: If we treat it as a classification problem, then only 301 vehicle-level labels are provided, and the labels are highly imbalanced (50/251). Furthermore, if we assign the 301 vehicle-level labels to snippets, there will be a significant mislabeling problem since even a failed battery system would have many normal charging snippets in an early stage (see the example in Appendix B). Therefore, a classification formulation did not generate a successful model.
>
> **Q**: Gap between Section 3 and Section 4.
> **A**: We realize that the writing transition is not very smooth, and now we have updated the first two paragraphs in Section 4. More specifically, we want to identify the likelihood $l^* \in \mathcal{L}$ that minimizes the KL divergence between the empirical distribution $\hat{p}\_{0}$ and the probability function $p_{l^*}$ induced by the learned likelihood $l^*$. Please refer to the updated paragraph for a more detailed description.
>
> **Q**: Why DyAD is more appealing than competing techniques?
> **A**: Now we have summarized the benefits where DyAD gains performance improvement in the last paragraph of Section 4. More specifically, existing algorithms are mostly built on the intuition that a model could learn generalizable normal patterns in the time series. DyAD improves this idea by
> 1) We propose a dynamical system probabilistic model which isolates rare input signals (e.g., battery charging protocol) from abnormal systems (e.g., battery status).
> 2) By focusing on hypothesis testing in the system parameter space $\theta$ as opposed to the time series itself, the dimension of the distribution to be learned is greatly reduced. Thanks for your suggestion and we have updated the writing. We hope you have time to read the updated part.

---

> ### Author Response · Authors · 2022-11-18
> **Further response to Reviewer 4Vb4**
>
> We are happy to answer any further questions and make edits according to suggestions.

---

### Official Review · Reviewer_R2Bu · 2022-10-28

**Confidence:** 4
**Correctness:** 3
**Technical Novelty And Significance:** 1
**Empirical Novelty And Significance:** 1
**Recommendation:** 1

**Clarity, Quality, Novelty And Reproducibility:**

The paper lacks novelty and originality. I am also not satisfied with the presentation of the work. I have difficulties identifying those parts of the work the authors treat as their original contributions. The data set's description is unclear to me since I am not an expert in this area. Also, the presentation of the results on the benchmarks in section 6 is insufficient to evaluate and reproduce the performance.

**Strength And Weaknesses:**

Time series anomaly detection is an important area nowadays to monitor dynamic systems (for example, weather) and to gain insights into unknown processes by analysing detected anomalies. This usually comes with a continuous time series in which intervals that show anomalous behaviour/relationship of the dynamic process variables involved have to be detected. However, this paper seems to treat anomaly detection as an outlier detection method in static data built from a time series model, i.e. the whole time series is input to the system to determine whether it shows anomalous behaviour. This observation is a major criticism since the paper title is - at least for me - misleading. However, such preconditions/assumptions might be of importance in some applications, like the introduced battery health data set. But even in this case, the paper lacks originality and contribution to the area. Another criticism is the strong assumption of the Gaussianity of the involved variables. I wonder whether Kalman-Filter with control would do a similar job with a proper evaluation function of the model match.

**Summary Of The Paper:**

The paper suggests a specific method for anomaly detection in time series data. The approach is based on hypothesis testing. Given the distribution of the normal data and the empirical one from test data, the decision of normal vs. abnormal is made. The authors suggest as an application area the evaluation of the battery health of e-cars and provide a new data set for future benchmarks in anomaly detection. In the experiments, they evaluate their method on such data. In addition, two public benchmarks are used to compare with existing methods. The experimental results seem to support the claims of the authors

**Summary Of The Review:**

If the contribution by the authors is the solution for the battery system anomaly detection (Sec. 5), then it boils down to Algorithm 1 (basically an averaging operation) and a variance test. This is not sufficient for me as the contribution, even if there might be more complex problems around that might be tackled by their method. Furthermore, the theoretical part of their paper assumes Gaussianity for the time series process variables (page 6, enumeration 1-3). I cannot see any novelty in this since the authors do not relate such dynamic systems to the Kalman filter.

---

> ### Author Response · Authors · 2022-11-14
> **Response to Reviewer R2Bu**
>
> Thank you for your time reviewing our paper!
>
> **Q**: “If the contribution by the authors is the solution for the battery system anomaly detection (Sec. 5), then it boils down to Algorithm 1 (basically an averaging operation) and a variance test.”
> **A**: We respectfully disagree with the statement, as Algorithm1 is only an intermediate step for applying our proposed model to the EV anomaly detection task. We clarify that
> 1. We formulate time series anomaly detection as hypothesis testing on a hidden Markov chain. Based on this framework, we propose a novel deep learning architecture. And the model is more sample efficient by exploiting the underlying probabilistic model.
> 2. Our algorithm can apply to both our released battery system and conventional time series anomaly detection tasks (two spacecraft datasets). In the first case, DyAD requires Algorithm 1 to compute the anomaly score. In the second case, DyAD can be applied directly without Algorithm 1, just as any other time-series anomaly detection algorithm.
>
>
> **Q**: Kalman Filter?
> **A**: Kalman Filter infers hidden states from observations. The common point between Kalman Filter and our framework is that we both consider a hidden Markov chain. However, the standard Kalman filter technique cannot be applied to anomaly detection for a few reasons:
> 1. Standard Kalman filter requires known system dynamics. Learning the dynamics from the data efficiently requires careful model design such as part of our proposed architecture.
> 2. Most efficient computational algorithms for Kalman filter cannot be applied to nonlinear systems. Integrating nonlinearity via neural networks is an open problem that is still being actively investigated by the community.
> 3. Even if states can be estimated via modeling observations through the Kalman filter, it’s unclear how to give anomaly detection from the estimated states.
>
> Furthermore, we argue that the gaussianity assumption is commonly used in both classical deep probabilistic models (e.g., variational autoencoder) and state-of-the-art models (e.g., denoised diffusion probabilistic model). Using Gaussian noise does not imply that the algorithm is similar to the Kalman filter.
>
> **Q**: This is an anomaly detection task or time series anomaly detection task?
> **A**: Our theoretical framework is built for the time series problem. Based on the framework, we propose DyAD. DyAD can be directly applied to existing time series anomaly detection datasets like MSL and SMAP. Both datasets are included in our paper. We also include a new experiment on SWaT dataset now, and the results are listed in the response for Reviewer V5eP. As for the EV dataset, in fact, it is a more challenging time series anomaly detection task. The anomaly labels are more sparse (labels can and only can be labeled on vehicle level) and limited (only 301 vehicles are available). We aim to detect future battery failure with data collected from an early stage to prevent potential damage to passengers (see an example in Appendix B).

---

> ### Author Response · Authors · 2022-11-18
> **Further response to Reviewer R2Bu**
>
> We are happy to answer any further questions and make edits according to suggestions.

---

### Official Review · Reviewer_V5eP · 2022-10-31

**Confidence:** 3
**Correctness:** 3
**Technical Novelty And Significance:** 2
**Empirical Novelty And Significance:** 2
**Recommendation:** 6

**Clarity, Quality, Novelty And Reproducibility:**

The paper is well-written and easy to follow. The authors make clear and concise statements about their motivation of the study. In my opinion, however, this is not a fresh idea that a well-formulated probabilistic model can make sample efficiency much higher and lead to better model design. The authors released the source code and an EV dataset. I believe the study is reproducible.

**Strength And Weaknesses:**

Strengths:
- The study demonstrates that a well-formulated probabilistic model can significantly increase sample efficiency and result in better model design.
- The dataset released by the authors could be useful for the community.
- The proposed model performed significantly better than the other candidate models.

Weaknesses:
- Properly formulating the probabilistic model underlying the data is not a brand-new concept. This is more prevalent in literature.
- Experiments are somewhat limited


**Summary Of The Paper:**

In this paper, the authors developed an auto-encoder-based anomaly detection model called DyAD (Dynamic system Anomaly Detection) for time series anomaly detection. They primarily dealt with the high-dimensionality issue caused via viewing time series anomaly detection as hypothesis testing on dynamical systems. They demonstrated the performance of their model on both public dataset and on a newly prepared EV dataset.

**Summary Of The Review:**

This is an interesting study for time series anomaly detection. I believe the research community may find value from this study. Even though the authors released the source code, it would be helpful if they could briefly discuss their implementation details in the manuscript (maybe in the appendix). Please add a comma (,) after e.g. (page 6).

The study demonstrated state-of-the-art results in several datasets, including an EV dataset and two spacecraft datasets. Although the authors claim that they validated the efficacy of their model using these datasets, I have a little reservation about the diversity of the datasets and the generalizability of the proposed model. I would like the authors to extend their experiments on one or more general purpose tasks, such as energy systems, stock analysis and meteorological data to evaluate the robustness of the model. We cannot draw conclusions about the robustness of a model based on limited experiments.

Overall, I am bit concerned about the generalizability of the model since the experiments are not conducted on a variety of datasets. I therefore do not recommend accepting the paper in the current form considering a high bar to ICLR.

---

> ### Author Response · Authors · 2022-11-14
> **Response to Reviewer V5eP**
>
> Thank you for your time reviewing our paper and for your valuable comments! We have included additional experiments as suggested.
>
> We also hope to comment that integrating probabilistic modeling into existing deep learning pipeline is still under active development by the community. Our work provides a successful application to time-series anomaly detection from theoretical analyses to real-world experiments. We believe that our theory framework (we failed to find in other deep anomaly-detection work) and new large-scale real-world dataset could be of the community’s benefit, and inspire new algorithm design.
>
> We are happy to answer any further questions.
>
> Details are as follows.
>
> **Q**: Formulating the probabilistic model underlying the data is not a brand-new concept?
> **A**: Indeed, a better probabilistic model can lead to better algorithm design. However, successfully combining probabilistic models with deep learning models is still an open problem. For example, classical probabilistic models are often linear in compromise for computability, whereas more modern models introduce nonlinearity by incorporating neural network structures. Researchers have made many attempts throughout the past years (Variational autoencoder 2013 [1], Normalizing flow 2015 [2], Diffusion probabilistic models 2015 [3], noise-conditioned score network 2019 [4], denoised diffusion models 2021 [5], etc.). In this work, we show that the idea can be successfully applied to time-series anomaly detection by proposing an effective and efficient implementation that can solve real-world problems.
>
> We further highlight that although there is an apparent underlying probabilistic model in the anomaly detection problem, it has not been formalized or explored in the multivariate time series setting.
>
>
> **Q**: Experiments are limited? Generalize to other datasets?
> **A**: We looked into the time-series anomaly detection literature and found that there is no consensus on large-scale datasets for benchmarking algorithms. This is also one motivation for us to release a clean, large-scale, real-world EV dataset.
>
> We picked the spacecraft datasets because they are the most popular ones with published implementations. Following the reviewer’s suggestion, we now implement DyAD and other baseline algorithms in the SWaT dataset. The SWaT dataset is collected from a water treatment test-bed which records sensor data. The results are shown in the following table. We can see that DyAD outperforms other baseline algorithms.
>
> | Algorithm | F1 | Precision | Recall |
> |---|---|---|---|
> | AE 		|0.7961| 0.9452| 0.6876|
> |DeepSVDD   |	0.7870 |0.8633 |0.7231|
> |LSTMAD 	|	0.8007 |0.9570| 0.6883|
> |MTAD-GAT 	|	0.8359 |0.9271 |0.7612|
> |GDN 		|	0.8082 |0.9935 |0.6812|
> |DyAD 	|	0.8399 |0.9824 |0.7335|
>
> We notice that the gap is not significant, but DyAD can consistently produce the best performances. If you are interested in a specific dataset, please let us know, and we will try our best to implement DyAD on it.
>
> **Q**: Implementation details?
> **A**: Due to limited space, the implementation details (training epochs, batch size, brief network architecture introduction, learning strategy) of DyAD and other baseline algorithms are discussed in Appendix C. Our supplementary material also contains all the code needed to reproduce the results.  Following the reviewer’s suggestion, we have also added additional details.
>
>
> [1] Kingma, Diederik P., and Max Welling. "Auto-encoding variational bayes." arXiv preprint arXiv:1312.6114 (2013).
> [2] Rezende, Danilo, and Shakir Mohamed. "Variational inference with normalizing flows." International conference on machine learning. PMLR, 2015.
> [3] Sohl-Dickstein, Jascha, et al. "Deep unsupervised learning using nonequilibrium thermodynamics." International Conference on Machine Learning. PMLR, 2015.
> [4] Song, Yang, and Stefano Ermon. "Generative modeling by estimating gradients of the data distribution." Advances in Neural Information Processing Systems 32 (2019).
> [5] Ho, Jonathan, Ajay Jain, and Pieter Abbeel. "Denoising diffusion probabilistic models." Advances in Neural Information Processing Systems 33 (2020): 6840-6851.

---

> > ### Comment · Reviewer_V5eP · 2022-11-18
> > **Acknowledgement on the authors' responses**
> >
> > I have read the authors' responses and other reviews. I appreciate the authors' efforts to disseminate their work to the community. They addressed most of my concerns. Despite the limited contribution of the study, I think it has value on its own. The community may also benefit from the dataset released by the authors. Overall, I recommend accepting the paper and raising my previous score accordingly.

---

### Decision · Program_Chairs · 2023-01-20

**Decision:**

Reject

**Justification For Why Not Higher Score:**

First of all, the presentation is not clear enough to highlight the main contribution.

**Justification For Why Not Lower Score:**

N/A

**Metareview: Summary, Strengths And Weaknesses:**

This paper presents an anomaly detection method in time series data, referred to as ‘DyAD’ that is based on the hypothesis testing on dynamical systems. The method leverages an autoencoder architecture for modeling the underlying dynamical system. All of reviewers agree that a new dataset for battery power of electric vehicles is valuable for community and the timely topic is tackled by a well-formulated probabilistic model. However, there are critical concerns that should be carefully considered for future submissions. The presentation should be improved. It is not clear which part is a new contribution since the approach in this paper is not a fresh idea. Sections 3 and 4 are not well connected. In some place, the notation is confusing. Since the use of an autoencoder is popular in anomaly detection, I would like to suggest the authors to improve Figure 1 to highlight how different the overall scheme proposed in this paper is from existing ones. Therefore, the paper is not recommended for acceptance in its current form. I hope authors found the review comments informative and can improve their paper by addressing these carefully in future submissions.